# CharAs-CBert: Character Assist Construction-Bert Sentence Representation Improving Sentiment Classification

**DOI:** 10.3390/s22135024

**Published:** 2022-07-03

**Authors:** Bo Chen, Weiming Peng, Jihua Song

**Affiliations:** School of Artificial Intelligence, Beijing Normal University, No. 19, Xinjiekouwai St., Haidian District, Beijing 100875, China; bochen@mail.bnu.edu.cn (B.C.); pengweiming@bnu.edu.cn (W.P.)

**Keywords:** sentence representation, sentiment classification, internal structure information, construction vector, character vector

## Abstract

In the process of semantic capture, traditional sentence representation methods tend to lose a lot of global and contextual semantics and ignore the internal structure information of words in sentences. To address these limitations, we propose a sentence representation method for character-assisted construction-Bert (CharAs-CBert) to improve the accuracy of sentiment text classification. First, based on the construction, a more effective construction vector is generated to distinguish the basic morphology of the sentence and reduce the ambiguity of the same word in different sentences. At the same time, it aims to strengthen the representation of salient words and effectively capture contextual semantics. Second, character feature vectors are introduced to explore the internal structure information of sentences and improve the representation ability of local and global semantics. Then, to make the sentence representation have better stability and robustness, character information, word information, and construction vectors are combined and used together for sentence representation. Finally, the evaluation and verification are carried out on various open-source baseline data such as ACL-14 and SemEval 2014 to demonstrate the validity and reliability of sentence representation, namely, the *F*_1_ and ACC are 87.54% and 92.88% on ACL14, respectively.

## 1. Introduction

As one of the important operations of natural language processing, sentence embedding representation is widely used in many tasks such as text classification [1], semantic matching [2], machine translation [3], and knowledge question answering [4]. The current popular sentence representation methods are mainly based on neural networks and pre-trained language models. The most widely used neural network models are long and short-term memory networks [5], as well as convolution [6] and attention models [7], etc. When processing a sentence, these neural network methods are in a smooth order, while the basic structure of the sentence is not considered, meanwhile, in the subsequent sentence synthesis process, the basic syntactic information of the sentence is ignored, such as the obvious difference between the synthesis of “adverb-noun” and “adjective-noun”. When using a simple pre-trained language model BERT [8] to achieve sentence representation, it is easy to lose a lot of sentence details, resulting in sentence representation tasks that are still lower than the traditional Glove word embedding representation method. Therefore, the Bert language model is usually combined with neural network methods. Improving sentence representation has become a new trend.

Unlike convolutional neural networks [6], which are limited by the receptive field, the attention mechanism [7] balances the interrelationships between words in a sentence using weight calculation and assignment, meanwhile, highlighting the representation of salient features. Recurrent neural networks cannot establish effective long-term dependencies in more complex sentence representation tasks due to problems such as gradient disappearance and explosion. The Independent recurrent neural network (IndRNN) [9] improves neuron performance because each neuron has independent spatiotemporal features. The interpretability of the input behavior and the cross-layer connection between them prevents the gradient from disappearing and exploding and promotes the long-term learning of the network to improve the modeling ability of dependencies. In the traditional language model, Bert needs a lot of time for clustering and similarity analysis of sentence pairs to realize the embedded representation of sentences. For example, the sentence-BERT [10] algorithm uses the average value of the BERT output vector for supervised learning to achieve efficient sentence embedding. However, when interpreting sentences, the same word may also be ambiguous in different sentences, that is, the role of a word in a sentence depends on the context of the sentence, not entirely on the word itself. In addition, when two sentences have the same construction, the phrases that make up the sentence may also be quite different. Such constructions as “Not only did they have amazing, sandwiches, soup, pizza, etc, but their homemade sorbets are out of this world!” and “I stumbled upon this restaurant on my way home from the subway.” can be expressed as “NN -1 NN -1 NP -1”, but the internal structure of the phrase represented by “NP” in the construction is very different, and the same sentence construction is used to represent different sentences to guide the two. The compositions of phrases or sentences is irrational. Where the words in the sentence “Not only did they have amazing, sandwiches, soup, pizza, etc, but their homemade sorbets are out of this world!” are “soup” (NN), “pizza” (NN), and “world” “(NP) . In the sentence “I stumbled upon this restaurant on my way home from the subway.” “NN” is “restaurant”, ‘way’, “home”; “NP” is “subway”, the rest are “−1” contains punctuation. Therefore, it is necessary to obtain the internal structure information of the words in the sentence to improve the ability of the construction information to distinguish the basic structure of the sentence.

To alleviate the ambiguity of the same word in different sentences and combine the construction to explore the internal structure information of the sentence, a better sentence representation can be obtained. We propose a sentence representation framework of character-assisted construction Bert to improve the representation of global and contextual semantic information in sentiment texts. The approach has three main contributions:When understanding a sentence, the weight of the word processed by BERT is not directly used to explain the sentence, but a slice attention enhancement network is designed to explain these behaviors, assigning salient words in the sentence to the sentence. Higher weight coefficients, meanwhile, explore the channel dependencies and spatial correlations of different salient words in the sentence.Based on our sentence construction, we design a bidirectional independent recurrent neural network to explore the construction vector of sentences, alleviate the ambiguity of the same word in different sentences, and promote long-term learning of the network to establish effective long-term dependencies, it realizes the interaction between forward and backward semantic information and improves the model’s ability to perceive contextual details.A construction-based character graph convolutional network is designed to explore the internal structural information of salient words in sentences, that is, there is a strong correlation between adjacent characters in each salient word. Features strengthen construction information to improve the ability of construction information to distinguish the basic structure of sentences. Furthermore, we design a triple loss function to better tune and optimize the network to learn better sentence representations.

The rest of this paper is organized as follows: Section 2 presents the related research work on sentence representation and sentiment classification; Section 3 details our proposed CharAs-CBert sentence representation framework and introduces the internal components of the framework in Section 3.1 and Section 3.2, respectively; experimental results and analysis are presented in Section 4; conclusions and future research plans are drawn in Section 5.

## 2. Related Works

Sentiment classification usually maps emotional sentences into a dense vector space and determines the author’s subjective attitude, i.e., positive, negative or neutral, by solving the sentence representation. How to improve the performance of sentence representation to obtain complete and accurate semantic information and improve the downstream task has been a widespread concern of researchers.

For several traditional machine learning algorithms, Bayhaqy et al. [11], respectively, used the methods of the decision tree, K-Nearest Neighbor (K-NN), and Naive Bayes classifiers to conduct experiments on the data of different users’ opinions on the Indonesian market on Twitter. Naive Bayes classifiers is more suitable for sentiment analysis in the market research category. Rathi et al. [12] analyzed various machine learning algorithms on Twitter comments, but the effect was not good. Finally, they proposed the method of combining the support vector machine (SVM) and decision trees through experiments. Anwar et al. [13] conducted emotion analysis on large-scale Steam Review data set, respectively, using a fully supervised machine learning decision tree and naive Bayes. higher than the naive Bayes algorithm, which benefited from the additional features of the data set to enhance the learning effect of the decision tree. Chang et al. [14] wrote the use of the library support vector machine (LIBSVM) of support vector machine (SVM) and gave many details of support vector machine (SVM) implementation so that more people can use SVM to realize sentence classification more conveniently. Li et al. [15], based on the library support vector machine (LIBSVM) library and Gaussian radial basis function kernel with kernel and cost parameters as default values experimented with support vector machine (SVM) and nearest neighbor algorithm (NN) to achieve emotion classification.

With the development of the machine learning model, the accuracy of the sentence representation model is gradually improved, but there is still the problem of incomplete sentence semantic expression. With the emergence of deep learning, the unique learning model of the neural network provides a distributed lexical expression for sentences, which further promotes the development of this field. Ma et al. [16] proposed a tree-dependent self-attention network by combining the self-attention mechanism with trees, considering the basic form of sentences, to explore the deep semantic information of sentences, but the network was not able to obtain features when acquiring features, relying too much on the basic structural information of the sentence, ignoring the role of some keywords in the sentence. Task-Related Sentence Representation Bai et al. [17] designed a neighbor attention sentence representation method, which explicitly pointed out the label space in the input and predicted the class of the label by adding mask labels while using the fusion label to obtain the sentence’s semantic information. However, the model is too complex and relies heavily on label information. Hu et al. [18] proposed a pruned trained recurrent neural network for grammar induction and text representation using a top-down parser as a pruning method for the model and parallel encoding during inference, but this method in the pruning process is easy to ignore the detailed semantics, which leads to the reduction in the representation accuracy of the model. Fu et al. [6] proposed a coding-decoder model with an attention mechanism. Convolution neural networks (CNN) were used as an encoder to learn distributed sentence representation features, and Long Short-Term Memory (LSTM) with an attention mechanism as a decoder could not only reconstruct the original sentence structure but also multi-task to predict the next sentence content. Good results are obtained in multiple datasets. Zhao et al. [19] use the attention mechanism to obtain semantic representation at different levels in sentences, which can more accurately express the emotion reflected in the text and make the sentence representation more comprehensive. Most of the neural network training models remain in the training of vocabulary targets, and there are few studies on the whole sentence as the training target. In this regard, Wu et al. [20] proposed the contrast learning (CLEAR) method represented by sentences, in which enhanced strategies such as deleting, replacing, and rearranging words in sentences were used so that the results achieved higher accuracy than other methods in SentEval and GLUE data sets.

Contrastive learning has made great progress in sentence representation. Zhang et al. [21] proposed a comparison model MixCSE on this basis, which improved sim-CSE by constructing a negation mechanism on feature learning of counterexample samples and had a positive effect on sample classification with similar semantics. At present, various scholars have proposed effective models for training and classification, but due to the lack of high-quality data sets, the training model has always failed to reach the ideal state. In this environment, unsupervised learning does not depend too much on the quality of data sets and has been widely studied in recent years. Zhang et al. [22] proposed an unsupervised sentence representation learning method based on Bootstrapped Sentence Representation Learning (BSL). Given an enhanced view of each sentence, the sentence was trained to adjust a branch of the network online, among other different sentence representations. This method is obviously due to other unsupervised methods, but it still has some shortcomings in sentence enhancement. Similarly, Xu et al. [23] also proposed an unsupervised multi-task framework (USR-MTL), which mixed multiple sentence learning tasks into the same framework and obtained more meaningful sentence representation input by learning word order, word prediction, and sentence order. The learning process could be completed by using an untagged corpus. To balance the role of different words, Seo et al. [24] divided sentences into the form of Token, and combined the attention mechanism and sentence BERT, aiming to assign corresponding weights to different words through the attention mechanism to highlight the role of keywords. This improves the accuracy of subsequent tasks, but in the sentence decomposition and representation stage, it is easy to ignore the internal structure information of words, which weakens the representation effect of global semantics.

## 3. CharAs-CBert Framework

To alleviate the ambiguity of the same word in different sentences and explore the internal structure information of salient words in sentences by combining construction information, we developed a sentence representation framework of character-assisted construction-Bert (CharAs-CBert) to improve the classification performance of downstream tasks such as emotion classification. The overall network structure of CharAs-CBert is shown in Figure 1.

The CharAs-CBert sentence representation framework is mainly composed of three parts, namely the BERT initial embedding module with character and construction information, the characters auxiliary module, and the downstream sentiment classification module. Among them, the BERT embedding module of characters and constructions aims to map words and the characters that make up words into a low-dimensional space according to the sentence construction, making it easier to represent sentences; the character auxiliary module uses characters graph convolution (CharGCM) starting from the smallest unit that constitutes a sentence, that is, characters, it explores the internal structural information of sentences and improves the representation ability of salient words. Based on the sentence construction, we model its long-term dependencies with the help of a bidirectional independent recurrent neural network (BIRM), and realize information interaction from forward and reverse; in addition, we utilize the slice attention module (SAM) from two perspectives, such as channel and space. Model the importance of each word in a sentence and assign a higher weight ratio to keywords. At the same time, considering that the prior knowledge contains rich underlying semantics, we introduce prior word vector knowledge to improve the sentence representation ability. SoftMax classifiers are designed to perform downstream tasks such as sentiment classification.

### 3.1. Initial Embedding Module

As the most basic components of a sentence, the representation effect of a word directly affects the representation of the entire sentence and the character is the smallest unit of a word, so obtaining an effective character vector is helpful for sentence representation. Therefore, to improve the representation of sentences, we use the pre-trained language model BERT [8,10] to embed the words and characters that make up sentences, as well as the construction information that distinguishes sentence forms, into a unified low-dimensional space, and obtain the global sum of sentences from different perspectives. Context semantics, as well as alleviating the ambiguity of the same word in different sentences, further explores the internal structure information of sentences through character vectors, highlighting the differences between different sentences under the same construction. It is worth noting that “construction” usually means “construction grammar”, which is a grammar theory that gradually emerged in the late 1980s and a research method and school adapted to almost the entire language category. Constructive grammar is born out of cognitive grammar, which is a rebellion against formal grammar. It belongs to the category of cognitive linguistics in essence, but it has the characteristics of being an independent paradigm of language research. In a certain sense, constructionism has formed an independent school of research, whose purpose is to emphasize inductive descriptions to explain existing sentences, rather than to generate possible legal sentences according to rule constraints, a certain expression construction always corresponds to a certain meaning.

Suppose that each sentiment sentence consists of *N* words, which is represented as $x=\{W_{1},\cdots ,W_{i},\cdots ,W_{N}\}$, and each word contains *M* characters. At the same time, the sentence may contain *S* different constructions, such as the sentence “But the staff was so horrible to us.” subjective attitude is “negative”, but it has “CC -1 NP -1 .” and “DT -1 NP -1 .” and other construction structures, so to obtain an effective sentence construction vector, improve the ability of semantics to distinguish sentence structure, and promote sentence representation, we adopt an average weighting strategy to count the words corresponding to “CC”, “NP” and “DT” in the construction and obtain the mean vector feature, where “CC” indicates coordinating conjunction; “DT” indicates determiner; and “NP” indicates noun phrase or noun; in the sentence “But the staff was so horrible to us.” “But” is used as a “connector (CC)” or “determiner (DT)” to the noun phrase “the staff” (NP). Limit the scope of the situation or connect with other words to form a specific state. It is worth noting that the words corresponding to “−1” in the construction are not calculated for the mean value. The character vector, word vector, and construction vector of the corresponding word in the sentence are shown in the following equation.
(1)$$\left\{\begin{array}{ll} f_{W_{i,j}}^{Ch}=BERT\left(W_{i,j}\right), & 0<j\leq len(W_{i})\\ f_{W_{i}}=BERT\left(W_{i}\right), & 0<i\leq len(x)\\ f_{W_{i}}^{Co}=\frac{1}{t+1}\sum_{t=0}^{T}BERT\left(W_{i}\right), & 0\leq T\leq S \end{array}\right.$$
where $f_{W_{i,j}}^{Ch}$ represents the vector information of the $j^{th}$ character in the $i^{th}$ word; $f_{W_{i}}$ represents the word vector feature of the $i^{th}$ word in the sentence; $f_{W_{i}}^{Co}$ represents the construction vector of the $i^{th}$ word. BERT(·) represents the pre-trained language model; *T* represents the number of constructions in the $i_{th}$ word $W_{i}$; when t=0, it means that the word corresponds to “−1” in the construction, and no mean calculation is performed. The initial representation of the sentence by feature vectors at different levels such as characters, words, and constructions is shown in the following equation.
(2)$$\left\{\begin{array}{l} f_{Ch}\left(x\right)=\left\{f_{W_{1,1\cdots M}},\cdots ,f_{W_{N,1\cdots M}}\right\}\\ f_{Wo}\left(x\right)=\left\{f_{W_{1}},\cdots ,f_{W_{N}}\right\}\\ f_{Co}\left(x\right)=\left\{f_{W_{1}}^{Co},\cdots ,f_{W_{N}}^{Co}\right\} \end{array}\right.$$

$f_{Ch}\left(x\right)$ indicates the character feature vector of the sentence; $f_{Co}\left(x\right)$ represents the construction feature vector of the sentence; $f_{Wo}\left(x\right)$ represents the word vector feature of the sentence; *x* represents the input sentence; $W_{i,1\cdots M}$ represents that each word in the sentence consists of *M* characters, and *M* in different words is different; *N* represents the number of words contained in the sentence.

A word is composed of several characters. We input the characters that make up the word into BERT for learning and training to obtain character features. Secondly, the construction of a sentence is also composed of specific words, and these specific words are input into BERT to obtain a vector feature. We think this vector feature is a construction vector that contains construction information.

This module explores the global and contextual semantics of sentences from different perspectives and levels by obtaining word vectors, character vectors, and construction vectors of emotional sentences. The character vector and construction vector can start from the internal structure of the words in the sentence and the basic structure of the sentence, effectively reducing the ambiguity of the same word in different sentences. Meanwhile, it is helpful for sentence representation.

### 3.2. Sliced Attention Module (SAM)

Although the word vector feature of the sentence is effectively obtained by using the BERT [8] language model, the importance of different words in the sentence is different. To highlight the representation ability of the keyword and the important role of the keyword, we designed a slice attention module. We model it from both channel and space perspectives to capture the channel dependencies and spatial correlations of words in a sentence, and combine them to achieve more effective word vector features, which is helpful for subsequent sentence representation.

For a given feature vector $f_{Wo}\left(x\right)$, we first slice it along the channel to obtain ψ slice groups, i.e., $f_{Wo}\left(x\right)=\left[f_{Wo}^{1}\left(x\right),\cdots ,f_{Wo}^{\psi }\left(x\right)\right]$, secondly, we input each slice group into the channel attention and spatial attention [7,25] components, obtain the semantic responses of the features and generate a corresponding significance coefficient for each slice group feature. The specific calculation process is shown as
(3)$$\left\{\begin{array}{l} f_{CA}=CAtt\left(f_{k}\left(x\right)\right)\\ f_{SA}=SAtt\left(f_{k}\left(x\right)\right)=\sigma \left(\Phi \cdot GN\left(f_{k}\left(x\right)+b\right)\right)\cdot f_{k}\left(x\right)\\ f_{k}\left(x\right)\in f_{Wo}\left(x\right),1\leq k\leq \psi \end{array}\right.$$
where CAtt(·) indicates the operation of channel attention; SAtt(·) indicates the operate of spatial attention; GN(⋯) indicates Group Norm operate; Φ indicates weight matrix; *b* indicates bias matrix; · means dot product; σ(·) indicates activation function of sigmoid.

Channel attention CAtt(·) uses a simple global average pooling layer to generate channel-level statistics to embed global words; unlike channel attention, spatial attention SAtt(·) focuses on where the informative parts are located, which is complementary to channel attention. Therefore, to simultaneously channel and spatially model the input information, *f* and *c* are aggregated together for the attention feature of this slice group. Specifically as shown as
(4)$$\left\{\begin{array}{l} f^{SA}\left(f_{k}\left(x\right)\right)=Cat\left(\left[f_{CA},f_{SA}\right]\right)\\ f^{SA}\left(x\right)=\left\{f^{SA}\left(f_{1}\left(x\right)\right),\cdots ,f^{SA}\left(f_{k}\left(x\right)\right),\cdots ,f^{SA}\left(f_{\psi }\left(x\right)\right)\right\} \end{array}\right.$$
where Cat(·) represents simple splicing and fusion.

### 3.3. Bidirectional Independent Recurrent Module (BIRM)

Considering that in the process of sentence extraction, the recurrent neural network is prone to problems such as gradient disappearance and explosion. At the same time, it is difficult to establish effective long-term dependencies, and it is difficult to explain the input behavior due to the entanglement of neurons. Although Long Short-Term Memory (LSTM) [5] can effectively solve a series of problems existing in recurrent neural networks [9], the use of hyperbolic tangent and sigmoid activation functions leads to hierarchical gradient decay, which cannot effectively capture the detailed features of sentences. Therefore, to address these limitations for more efficient sentence representations, we design a bidirectional independent recurrent neural network module (BIRM), which models the construction information from both forward and backward directions and realize the interaction of different directional features, which helps to obtain global and contextual details. In addition, the neurons in the same layer in this module are independent of each other, which is beneficial to the interpretation of input behavior. At the same time, the information flow is realized between different layers. Cross-layer connections help to model detail semantics and better transfer detail semantics to lower layers. The specific description of BIRM is as follows.
(5)$$\left\{\begin{array}{l} \vec{h_{z}}=\sigma \left(\vec{W}f_{Co}\left(x\right)+\vec{U}⊙\vec{h_{z-1}}+\vec{b}\right)\\ \overset{\leftarrow }{h_{z}}=\sigma \left(\overset{\leftarrow }{W}f_{Co}\left(x\right)+\overset{\leftarrow }{U}⊙\overset{\leftarrow }{h_{z-1}}+\overset{\leftarrow }{b}\right) \end{array}\right.$$

Among them, $\vec{W}$ and $\overset{\leftarrow }{W}$ represent the weight matrix of the forward and backward layers; ⊙ represents the Hadamard product. $\vec{h_{z-1}}$ and $\overset{\leftarrow }{h_{z-1}}$ represents the output feature of the z-1 layer in the forward-backward direction; represents the bias matrix; σ(·) represents the activation function of ReLu; each neuron in the network layer is independent of each other, and the connection between neurons can be achieved by stacking multiple layers. For the $n^{th}$ neuron, the output features of the front and rear directions are as follows
(6)$$\left\{\begin{array}{l} \vec{h_{n,z}}=\sigma \left(\vec{W_{n}}f_{Co}\left(x\right)+\vec{U_{n}}⊙\vec{h_{n,z-1}}+\vec{b_{n}}\right)\\ \overset{\leftarrow }{h_{n,z}}=\sigma \left(\overset{\leftarrow }{W_{n}}f_{Co}\left(x\right)+\overset{\leftarrow }{U_{n}}⊙\overset{\leftarrow }{h_{n,z-1}}+\overset{\leftarrow }{b_{n}}\right) \end{array}\right.$$
where $W_{n}$ and $U_{n}$ represent the input weight matrix and the circular weight matrix of the $n^{th}$ layer, respectively.

Each neuron in BIRM only accepts information from the input and its hidden state at the previous time step. That is, each neuron in BIRM independently processes a spatiotemporal pattern, i.e., independently aggregates spatiotemporal detail features over time by u. In addition, to obtain more efficient feature representations and avoid the catastrophic forgetting problem when the time is long, we use the residual method to connect multiple bidirectional independent recurrent modules. At the same time, to model the input construction information in two directions, such as forward and reverse, to improve the representation of contextual semantics, we fuse them. The fusion $f^{BIRM}\left(x\right)$ is shown as
(7)$$f^{BIRM}\left(x\right)=Cat\left(\left[\vec{h_{n,z-1}},\overset{\leftarrow }{h_{n,z-1}}\right]\right)$$
where Cat(·) represents the splicing operation.

### 3.4. Characters Graph Convolution Module (CharGCM)

Characters are the smallest units that make up a word, and the adjacent relationship between them reflects the internal structural information of the word. A sentence usually contains multiple constructions, and the constructions of different sentences may also be the same. When the constructions of different sentences are the same, it is not conducive to distinguishing between different sentences. Therefore, obtaining character information of words in a sentence is helpful for sentences. The distinction between representation and basic structure. For example, the subjective attitude of the sentence “We would return to this place again!” is “positive”, and the subjective attitude of the sentence “Too bad the food wasn’t of the same heritage.” is “negative”, but they have the same construction “DT−1NN.”. At the same time, there are also great differences in the length and composition of the phrases corresponding to “DT” and “NN” in the construction, and it may be better to use character information to distinguish these phrases from the inside. Where “DT” means determiner; “NN” means Noun, singular or mass, etc. Therefore, we construct a topological graph of a character, and use graph convolution [26] to optimize and adjust this topological graph, namely the character graph convolution module (CharGCM), to ensure the accuracy of the internal structure information of keywords.

Assuming that each word *W* in the input sentence *x* consists of *M* characters and can be expressed as $W_{i}=\left\{W_{i,1},\cdots ,W_{i,j},\cdots ,W_{i,M}\right\}$; The initial vector features of characters are obtained through the pre-trained language model BERT $f_{W_{i,j}}^{Ch}$, where *j* represents the $j^{th}$ character in the $i^{th}$ word; it is worth noting that there are 96 characters in total, including 52 uppercase and lowercase English letters, as well as 10 numbers and other special symbols. The overall characters can be “abcdefghijklmnopqrstuwxyzABCDEFGHIJ+=−<>()∗[]” etc.

Then, we construct a topology graph ζ from these *M* characters, such as the word “place” in the sentence “We would return to this place again!” which consists of five different characters. Each character can be considered as a graph node ν, the topology graph ζ is formally represented as ζ=(ν,ε), where $\nu =\{\nu_{1},\cdots ,\nu_{M}\}$ represents a set of character nodes; $\varepsilon =\{\varepsilon_{1},\cdots ,\varepsilon_{G}\}$, represents the set of edges between nodes, that is, the correlation between characters, *g* represents the number of edges; the adjacency matrix $A_{mp}$ of the topology graph ζ is shown as
(8)$$A_{mp}=\left\{\begin{array}{ll} \frac{f_{\nu_{m}}\cdot f_{\nu_{p}}^{T}}{2} & \nu_{m}\neq \nu_{p}\\ 1 & \nu_{m}=\nu_{p}\\ 0 & otherwise \end{array}\right.$$
where $\nu_{m},\nu_{p}\in \nu =\left\{\nu_{1},\cdots ,\nu_{M}\right\}$, $f_{\nu_{p}}^{T}$ indicates the feature information transpose of the $p^{th}$ node; $A_{mp}=0$ indicates that there is no edge weight between character nodes, that is, there is no correlation between these two adjacent characters. $A_{mp}=1$ indicates that the node is a self-looping node.

To obtain better character representations, we use two layers of graph convolutions to aggregate and transfer this graph node information. In the process of transfer and aggregation, to prevent the network from falling into local optimality, Laplace renormalization is introduced to operate the adjacency matrix *A*, and then, to obtain the character node information, the optimization process of the topology map ζ is as follows
(9)$$\left\{\begin{array}{l} f^{Ch}\left(x\right)=\sigma \left({\tilde{D}}^{-\frac{1}{2}}\tilde{A}{\tilde{D}}^{-\frac{1}{2}}x^{Ch}\Theta \right)\\ \tilde{A}=A+I \end{array}\right.$$
where $\tilde{D}$ denotes the degree matrix; *I* denotes the unit adjacency matrix; Θ denotes the weight matrix; $\tilde{A}$ denotes the renormalized adjacency matrix; $f^{Ch}\left(x\right)$ denotes the character feature after the graph convolution operation; σ(·) represents the activation function of “LeakyReLu”.

To improve the representation accuracy of sentences, we fuse character features, word vector features, and constructional information for sentence representation, and in addition, considering that the prior knowledge contains rich low-level semantic details, we embed it into the sentence representation. The specific operations are as follows.
(10)$$\left\{\begin{array}{l} u=Cat(\left[f_{A}^{Ch}⊗f_{A}^{SA},f_{A}^{Ch}⊗f_{A}^{BIRM},f_{p}^{A}\right])\\ v=Cat(\left[f_{B}^{Ch}⊗f_{B}^{SA},f_{B}^{Ch}⊗f_{B}^{BIRM},f_{p}^{B}\right])\\ f_{p}^{A}=MLP\left(f_{Wo}^{A}\left(x\right)\right),f_{p}^{B}=MLP\left(f_{Wo}^{B}\left(x\right)\right) \end{array}\right.$$

Among them, $f_{A}^{Ch}$ and $f_{B}^{Ch}$ represent the character vectors of sentences A and B, respectively; $f_{A}^{SA}$ and $f_{B}^{SA}$ represent the word vector features of sentences A and B, respectively; $f_{A}^{BIRM}$ and $f_{B}^{BIRM}$ represent the construction vector features of sentences A and B, respectively; $f_{p}^{A}$ and $f_{p}^{B}$ represent the prior knowledge of sentences A and B, respectively; ⊗ represent element-wise product; Cat(⋯) represent the connection operation; MLP(⋯) represent the multilayer perceptron operation.

Finally, the sentence vectors generated after fusion are applied to the downstream sentiment classification task to demonstrate the effect of sentence representation. Specifically as shown as
(11)$$\left\{\begin{array}{l} O_{s}=\left(u,v|u-v|\right)\\ O=SoftMax(O_{s}) \end{array}\right.$$

Among them, SoftMax(·) represents the classifier; ∣·∣ represents the similarity or distance.

To promote the proposed CharAs-CBert framework to obtain better sentence representation and alleviate the error caused by data imbalance, we embed class weights in the loss function, and the specific operations are as follows
(12)$$\left\{\begin{array}{l} \tau_{total}=\alpha \tau_{fl}\left(CW\right)+\beta \tau_{mce}\left(CW\right)\\ \alpha +\beta =1 \end{array}\right.$$
where CW indicates class weights; α,β indicates learning factors; $\tau_{fl}\left(\cdot \right)$ indicates focal loss; $\tau_{mce}\left(\cdot \right)$ indicates multiclass cross entropy loss.

## 4. Experimental Results and Analysis

In this section, we conduct rich experiments to demonstrate the effectiveness of the proposed CharAs-CBert framework in sentence representation and provide a detailed analysis of the experiments.

### 4.1. Datasets Preparation

**SemEval2014**: This dataset is mainly used for fine-grained sentiment classification, including two domains, Laptop and Restaurant, where each domain involves three categories: “positive”, “negative” and “neutral”, at the same time, the dataset of each domain is divided into training samples, validation samples, and test samples.

**ACL14**: The dataset mainly includes negative, neutral, and positive reviews of celebrity ties, products, and companies, with negative, neutral, and positive sentiment sentences accounting for 25%, 50%, 25%, respectively. Meanwhile, the number of training samples is 6248, and the number of test samples is 692. It is worth noting that each sentiment sentence in these datasets contains multiple different constructions, and Table 1 presents the statistical results of these data.

To demonstrate the effectiveness of the CharAs-CBert sentence representation framework and ensure the smooth progress of subsequent experiments, accuracy (*ACC*) and *F*_1_ values were adopted as evaluation indexes, as shown in the equation
(13)$$\left\{\begin{array}{l} ACC=\frac{TP+TN}{TP+FP+FN+TN}\\ F_{1}=\frac{2TP}{2TP+FN+FP} \end{array}\right.$$
where *TP* (true positive) indicates that the sample originally belongs to the positive class is divided into positive classes. *TN* (true negative) indicates that the sample originally belonged to the negative category and is divided into the negative category. *FP* (false positive) indicates that errors that originally belong to the negative class are divided into positive classes. *FN* (false negative) indicates that the error that originally belonged to the positive class is classified into the negative class.

### 4.2. Parameters Settings

**Model parameters**. In the process of training, some important parameters of the details are as follows: (i) the initial vector set to 3 × $10^{4}$, and using the cosine annealing vector (“CosineAnnealingWarmRestarts(·)”) to adjust CharAs-CBert sentence representation framework; (ii) the optimizer is “AdamW”; (iii) the number of iterations is 500, and the batch processing is set to 32. In addition, to prevent the model from falling into the local optimal state, the bit loss rate in the CharAs-CBert framework is set to 0.25. In the Bi-IndRNN module, we set the initial neural unit to 300, which is the same as the embedded word vector dimension. In the char-GCNs module, the initial number of graph nodes is set to 16 and the number of layers of graph convolution is set to 2.

**Environment configuration**. The paper uses the Pytorch1.7.0+cu110 platform to implement the CharAs-CBert model. All the code is developed based on python3.7. Meanwhile, to ensure the fairness and correctness of the experiment, all the experiments are carried out on two RTX3090
GPU cards.

### 4.3. Comparison with Other Models

To demonstrate the effectiveness of the proposed CharAs-CBert sentence representation framework, we conducted evaluation and verification on open-source baseline data such as SemEval2014 and ACL14. Table 2 shows the experimental results of different sentence representation methods.

According to Table 1, we can draw the following conclusions:The overall performance of our proposed CharAs-CBERT sentence representation framework on the three baseline datasets outperforms other representation models, such as $F_{1}$ in Laptop, Restaurant, and ACL14 than SBERT-att 1.1%, 1.03% and 1.17%, respectively. There are three possible reasons. First, we use the Slice Attention Module (SAM) to establish long-term dependent salient word representations from two directions, such as channel and space. The performance of sentence representation; second, BIRM and CharGCM are introduced to support construction information, explore the internal structure information of sentences, and highlight the differences between different sentences, resulting in the improvement of sentence representation performance; third, the fusion of three different feature vectors make up for the insufficiency of a single representation and understand sentences from different angles and levels. In addition, the introduction of rich low-level semantics further enhances the difference between sentences, improves the performance of sentence representation, and improves the downstream emotion. Accuracy for classification tasks.Compared with BERTS-LSTM and Tree-LSTM sentence representation models, TG-HTreeLSTM and TE-DCNN have certain competitive advantages in three types of data. For example, on the Laptop data, the $F_{1}$ of TG-HTreeLSTM is 6.53% higher than that of Tree-LSTM. The possible reason is that Tree-LSTM can only process binary selection trees that are different from the original selection tree. Conversely, TG-HTreeLSTM can process the original constituency tree of sentences, resulting in a performance improvement. The good performance of TE-DCNN may be because its dynamic synthesis strategy plays an important role, resulting in better semantic information obtained by the network.Capsule-B is improved by 0.98%, 1.42%, and 2.01%, respectively, compared with $F_{1}$ of TE-DCNN. The possible reason for this is that the capsule network can perceive more effectively due to the directionality of capsule neurons. The subtle changes between different sentences improve the distinguishing ability of sentence structure, thereby improving the representation effect of sentences.In the ACL14 baseline data, the ACC of CNN-LSTM is 1.09% higher than that of the LSTM method. The possible reason is that CNN obtains the local spatial features of sentences, LSTM encodes the time series and establishes a complementary relationship between the spatial and temporal features. Improved sentence representation. Thus, the sentence representation accuracy is improved. In contrast, Self-Att achieves better competitive advantages in three sets of open-source baseline datasets, mainly since self-attention focuses on key information and effectively models the local and global semantics of sentences.

To intuitively understand the operating efficiency of different sentence representation models, we give the parameters of different models and the visualization effect of FLOPs, as shown in Figure 2.

According to Figure 1, we can find that the Capsule-B sentence representation method has the largest FLOPs and Params. The main reason is that the capsule neuron is a vector neuron, which requires a large number of parameters to participate in the operation in the calculation process to ensure its representation performance. The FLOPs and Params of CharAs-CBert we mentioned are not optimal, but within the acceptable range, the framework has the best representation performance.

### 4.4. Ablation Studies

#### 4.4.1. Different Components of CharAs-CBert

To demonstrate whether the components in the proposed CharAs-CBert sentence representation framework play a positive role in the overall performance of the model, we evaluate and verify different components on three types of data. Table 3 shows the representation results of different components.

According to Table 3, we draw the following conclusions:Compared with the single-structure sentence representation, the multi-feature co-representation method shows better performance. Such as CharAs-CBert (SAM+CharGCM) vs. CharAs-CBert (CharGCM), CharAs-CBert (BIRM) and CharAs-CBert (SAM) $F_{1}$ on ACL14 baseline datasets increased by 1.27%, 1.78% and 2.02%, respectively. The possible reason is that the multi-feature vector fusion understands the sentence from different angles, and the different feature vectors form complementarity, making up for a single feature vector that is easy to ignore a question of detail semantics. In addition, the CharAs-CBert (BILSTM) method is inferior to the CharAs-CBert (BIRM) method on the three sets of open-source baseline data, which indicates that the proposed BIRM plays a positive role in the overall performance of the model. A possible reason is that stacking multiple layers of bidirectional independent recurrent neural networks obtains a better global representation.On the Laptop baseline data, CharAs-CBert (Att+BIRM+CharGCM) is better than CharAs-CBert (SAM+RNN+CharGCM) and CharAs-CBert (SAM+BIRM+CharCNN) $F_{1}$ is improved by 0.1% and 0.27%, respectively, which shows that our proposed components play a positive role in the overall performance of the model. In addition, we also found that the SAM component has the least positive effect on the model. It may be that the model only uses the word vector to represent the sentence in the absence of construction, ignoring the basic structure of the sentence, and cannot fully obtain the context of the sentence.Although SAM+BILSTM+CharGCM has achieved a certain competitive advantage, it is still lower than CharAs-Bert. Building deep BILSTMs for learning key semantics in data is difficult. In contrast, BIRM can be stacked into very deep networks using non-saturating activation functions, etc. We obtain better depth semantics and perceive richer detail changes due to stacking in the form of residuals.

#### 4.4.2. Comparing with Loss Functions

To verify whether the loss function we designed has a positive effect on the CharAs-CBert frameworks and the impact of the class weight (CW) on the model performance. The experimental results are shown in Figure 3.

As can be seen in Figure 3, the sentence representation using class weights (CW) is better than other loss functions without class weights. For example, $F_{1}$ and ACC of $\tau_{fl}\left(CW\right)$ are 1.11% and 0.09% higher than $\tau_{fl}$, respectively. The possible reason is that the class weights CW better capture the detailed semantics of few-shot classes. $\tau_{mce}\left(CW\right)+\tau_{fl}\left(CW\right)$ is 0.1%, which shows that the class weight CW is beneficial to the proposed CharAs-CBert framework to learn better sentence representations.

#### 4.4.3. Comparison of Different Layers

To demonstrate the influence of the number of layers in BIRM and CharGCM on the overall performance of the model, we conduct evaluation and verification on the Restaurant baseline datasets. Figure 4 shows the experimental results of different layers.

From Figure 4, we can see that with the increase in the number of layers in the BIRM and CharGCM components, the sentence representation effects ($F_{1}$) and Loss show a trend of increasing first and then decreasing. When the number of layers of BIRM is five and the number of layers of CharGCM is two, the sentence representation effect is optimal. For example, when the number of layers of CharGCM is two, the accuracy is higher than that of three layers. The possible reason for this is that a small number of layers cannot fully capture the contextual global semantics of a sentence, and when the number of layers is large, it is easy to reuse redundant information, resulting in a decrease in the accuracy of sentence representation.

## 5. Conclusions

In this paper, we design a new character-assisted structure BERT sentence representation framework, which utilizes words, structures, and characters to explore the context and global semantics of sentences from different perspectives and helps to capture the sentence’s meaning based on structure and character information. The internal structure information improves the ability to distinguish between different sentences. At the same time, due to the complementary and interactive relationship between different feature vectors, the ambiguity of the same word in different sentences is reduced. Finally, the evaluation results on baseline data such as ACL14 and SemEval 2014 show that the proposed CharAs-CBert sentence representation framework has good robustness and effectiveness, that is, the experimental results on different baseline datasets are superior to other sentence display methods.

Although the proposed CharAs-CBert sentence representation framework has achieved good representation performance, we found that this method still has shortcomings during the experiment, such as the complex model structure, and different feature vectors having some redundant information in the fusion stage. At the same time, there is still a lot of room for improvement in time efficiency. Therefore, in the following, we will design an efficient and concise semantic-guided sentence representation framework from the perspective of model structure and exploration of more effective detail semantics to obtain semantic details while ensuring accuracy and improving model efficiency.

## Figures and Tables

**Figure 1 sensors-22-05024-f001:**
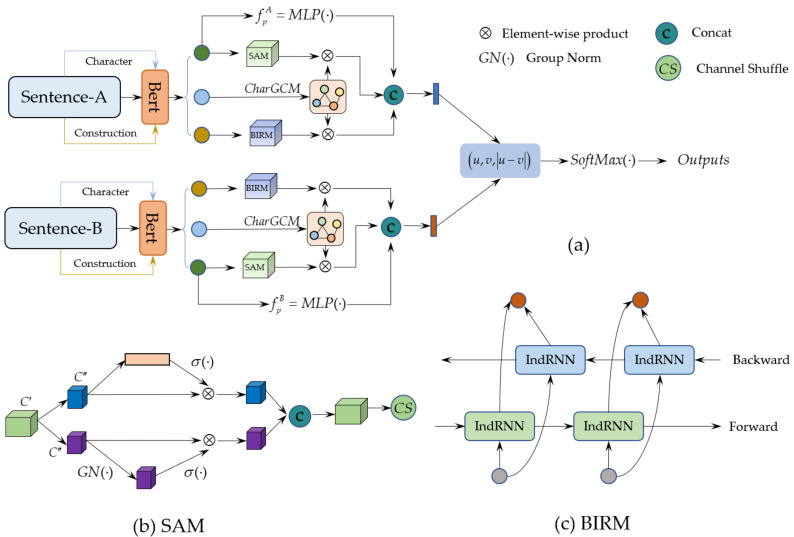
(**a**) The overall network structure of CharAs-CBert; (**b**) The slice attention module (SAM); (**c**) The bidirectional independent recurrent neural network module (BIRM). Where “CharGCM” indicates character graph convolution module; IndRNN indicates independent recurrent neural network; SoftMax(·) indicates classifier with softmax; σ(·) indicates the activation function of sigmoid; MLP(·) indicates the operation of multilayer perceptron; $C^{\prime }$ and $C^{\prime \prime }$ indicate the channels of input features and satisfies $C^{\prime \prime }=\frac{C^{\prime }}{2}$. GN(·) indicates the operation of GroupNorm; ⊕ indicates element-wise product operation. BERT indicates language model of pre-training; CS indicates channel shuffle operation; Backward indicates backward layers of BiIndRNN; Forward indicates forward layers of BiIndRNN. $f_{p}^{A}$ and $f_{p}^{B}$ represent the a priori word vector feature obtained by the multi-layer perceptron (MLP(·)) of sentence A and sentence B, respectively.

**Figure 2 sensors-22-05024-f002:**
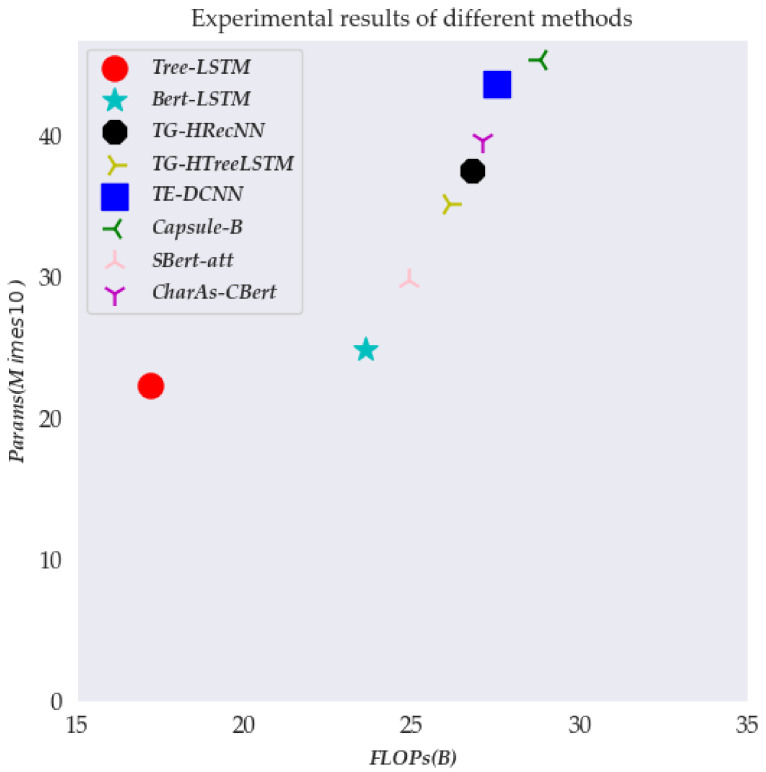
The performance of different sentence representation models. Params indicates the number of model parameters; FLOPs indicates floating-point operations.

**Figure 3 sensors-22-05024-f003:**
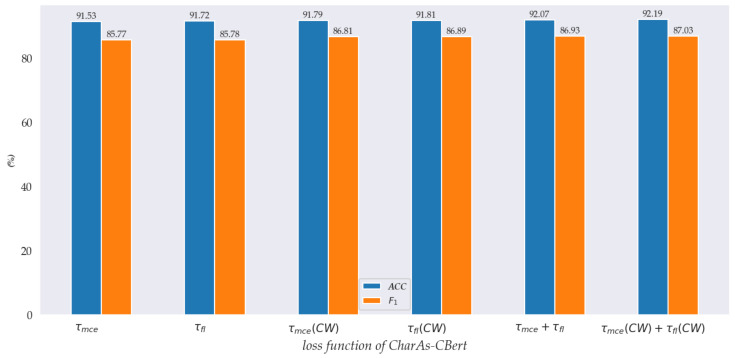
The performance of different loss functions on the Laptop datasets. CW indicates class weights; $\tau_{mce}$ indicates multiclass cross entropy loss; $\tau_{fl}$ indicates focal loss; $\tau_{mce}\left(CW\right)$ indicates multiclass cross entropy loss with class weights.

**Figure 4 sensors-22-05024-f004:**
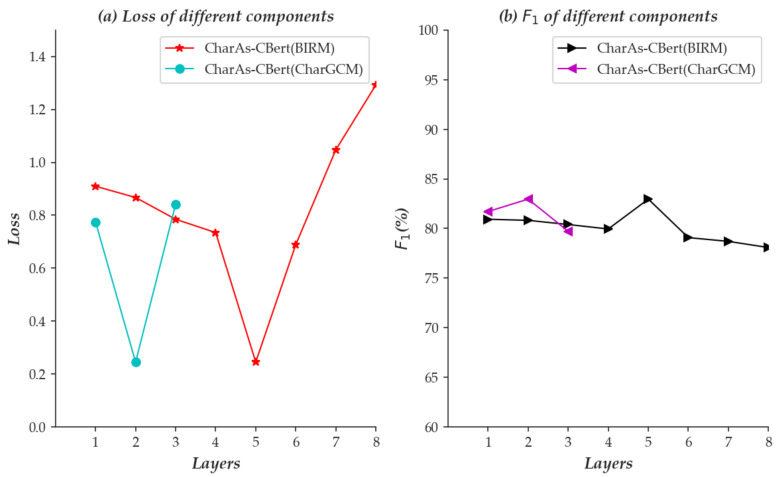
The performance of different layers with BIRM and CharGCM on the Restaurant datasets. (**a**) indicates loss performance; (**b**) indicates $F_{1}$ performance of different layers. Loss indicates the loss value of different layers. Layers indicates the number of BIRM and CharGCM.

**Table 1 sensors-22-05024-t001:** Statistical results of SemEval2014 and ACL-14 baseline datasets. “Consnum” indicates the numbers of sentence construction. “charnodes” indicates the number of characters nodes used to build a topological graph.

Datasets	Restaurant		Laptop		ACL14	
	Training	Testing	Training	Testing	Training	Testing
Positive	2164	728	994	341	3142	346
Negative	807	196	870	128	1562	173
Neutral	637	196	464	169	1562	173
Consnum charnodes	100,043 12,891	1,105,665 67,235	241,546 15,213	992,438 37,539	819,242 31,028	286,552 15,652

**Table 2 sensors-22-05024-t002:** Experimental results of different sentence representation methods. SemEval2014 including Laptop and Restaurant.

Model-Datasets	Laptop		Restaurant		ACL14	
	ACC	$F_{1}$	ACC	$F_{1}$	ACC	$F_{1}$
LSTM	75.38	72.24	73.98	70.07	77.42	73.19
CNN-LSTM	76.51	73.02	74.21	70.56	78.51	74.23
Tree-LSTM	78.08	74.88	76.64	72.89	80.5	77.06
BERT-LSTM	80.92	76.73	80.48	74.9	81.54	77.96
TG-HRecNN	82.08	79.52	80.93	75.92	82.46	80.63
TG-HTreeLSTM	83.03	81.41	80.96	76.42	85.83	82.17
TE-DCNN	87.55	83.25	83.93	78.99	87.49	83.84
Capsule-B	88.32	84.23	85.09	80.41	91.38	85.85
Self-Att [16]	86.51	82.42	83.79	78.64	86.92	82.74
SBERT-att [24]	90.59	85.93	85.31	81.93	91.53	86.37
CharAs-CBert	92.19	87.03	86.22	82.96	92.88	87.54

**Table 3 sensors-22-05024-t003:** Experimental results of different components. SemEval2014 including Laptop and Restaurant. SAM+BIRM+CharCNN indicates that the framework uses the three components SAM, BIRM, and CharCNN at the same time, and uses CharCNN to replace the CharGCM component we proposed; SAM indicates that the SAM component is used; BIRM indicates that the BIRM component is used; CharGCM means using CharGCM components; SAM+RNN+CharGCM means using RNN to replace our proposed BIRM components; Att+BIRM+CharGCM means using Att to replace our designed SAM components. SAM+BILSTM+CharGCM indicates using a bidirectional long-short memory network (BILSTM) to replace the BIRM.

Model-Datasets	Laptop		Restaurant		ACL14	
	ACC	$F_{1}$	ACC	$F_{1}$	ACC	$F_{1}$
CharAs-CBert (SAM)	86.35	82.4	81.34	77.65	86.01	83.48
CharAs-CBert (BILSTM)	87.12	82.33	82.05	77.79	87.54	83.65
CharAs-CBert (BIRM)	88.04	82.82	82.59	78.73	88.02	83.72
CharAs-CBert (CharGCM)	88.62	83.5	83.07	79.53	88.08	84.13
CharAs-CBert (SAM+CharGCM)	88.84	83.62	83.29	80.06	88.4	85.5
CharAs-CBert (SAM+BIRM)	89.85	84.17	83.43	80.65	88.64	85.54
CharAs-CBert (BIRM+CharGCM)	90.02	84.75	83.97	80.67	88.72	85.92
CharAs-CBert (SAM+BIRM+CharCNN)	90.03	85.48	84.24	81.04	89.62	86.32
CharAs-CBert (SAM+RNN+CharGCM)	91.3	85.65	84.67	82.03	90.62	86.42
CharAs-CBert (SAM+BILSTM+CharGCM)	91.47	85.69	84.92	82.19	91.14	86.71
CharAs-CBert (Att+BIRM+CharGCM)	91.96	85.75	85.44	82.56	92.27	86.92
CharAs-CBert	92.19	87.03	86.22	82.96	92.88	87.54

## Data Availability

Not applicable.

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
