# Peer review of "CharAs-CBert: Character Assist Construction-Bert Sentence Representation Improving Sentiment Classification"

_sensors, 2022, doi:10.3390/s22135024_

Round 1

Reviewer 1 Report

The language model proposed in [8] is usually referred as “BERT,” not “Bert.”

The “sentence construction” used here is not explained well. The same notation such as “NP -1 NN -1” is only found in a paper by the same authors.

A)      Chen, B., Peng, W., & Song, J. (2022). ConAs-GRNs: Sentiment Classification with Construction-Assisted Multi-Scale Graph Reasoning Networks. Electronics, 11(12), 1825.

I could not understand the example shown in Line 53 “NN -1 NN -1 NP -1”; how these tokens correspond to the previous two sentences?

Figure 1 (a): The organization of the network is similar to the Sentence-BERT, which has a Siamese-network topology (Figure 1 and 2 of [10]), but these networks are used when training the Sentence-BERT using the STS task that compares two sentences. The Sentence-BERT itself is not a Siamese network but a BERT model with pooling. On the other hand, the proposed CharAs-CBert network looks to receive one sentence and determines its sentiment. Why the proposed model takes two sentences?

Section 3.1: It is explained that the proposed network first takes words, characters, and constructions of a sentence and embed them using a pre-trained BERT. However, a usual BERT model only takes q sequence of words (or tokens) as an input. How the authors input characters and constructions into the BERT?

Line 140: The abbreviated word BSL is not defined elsewhere.

Table 2: Results of many conventional(?) methods are shown here, but none of those methods are explained. Thus, I could judge that the proposed method is truly better than the other methods.

The experimental result for the same dataset is shown in the literature A) above, which was just published by the same authors. I noticed that the absolute performance is different. It is natural that the methods are different, but I wonder why these two papers employed completely different set of recognition methods for the same task.

Section 3.3: One of motivations to introduce BIRM looks like that the tanh activation could be improved by ReLU activation. Changing the activation of LSTM is examined in the following paper:

B)      Kent, D., & Salem, F. (2019, August). Performance of three slim variants of the long short-term memory (LSTM) layer. In 2019 IEEE 62nd International midwest Symposium on circuits and Systems (MWSCAS) (pp. 307-310).

According to this work, changing the activation function from tanh did not improve the performance. Thus, the authors need to validate whether changing the activation improved the result compared with BLSTM. Table 2 shows comparison with many other models, but it lacks viewpoints from which the authors prove that the changes employed in the proposed method were effective.

Author Response

Point 1: The language model proposed in [8] is usually referred as “BERT,” not “Bert.”

Response 1: Thank you very much for the experts' opinions. We have made detailed modifications to this problem. 

For instance, when using a simple pre-trained language model BERT [8] to achieve sentence representation, it is easy to lose a lot of sentence details, resulting in sentence representation tasks that are still lower than the traditional Glove word embedding representation method. Therefore, the Bert language model is usually combined with neural network methods. Improving sentence representation has become a new trend.

Point 2: The “sentence construction” used here is not explained well.

Response 2: Thank you very much for the experts' opinions. We have made detailed modifications to this problem. 

For instance, As the most basic components of a sentence, the representation effect of a word directly affects the representation of the entire sentence, and the character is the smallest unit of a word, and obtaining an effective character vector is helpful for sentence representation. Therefore, to improve the representation of sentences, we use the pre-trained language model BERT [8][10] to embed the words and characters that make up sentences, as well as the construction information that distinguishes sentence forms, into a unified low-dimensional space, and obtain the global sum of sentences from different perspectives. Context semantics, as well as alleviating the ambiguity of the same word in different sentences, meanwhile, further explore the internal structure information of sentences through character vectors, highlighting the differences between different sentences under the same construction. It is worth noting that "construction" usually means  "construction grammar" and it is intended to emphasize that inductive descriptions explain existing sentences rather than generate possible legal sentences based on rule constraints.

Point 3: I could not understand the example shown in Line 53 “NN -1 NN -1 NP -1”; how these tokens correspond to the previous two sentences?

Response 3: Thank you very much for the experts' opinions. We have made detailed modifications to this problem. 

For instance,

such constructions as "Not only did they have amazing, sandwiches, soup, pizza, etc, but their homemade sorbets are out of this world!" and "I stumbled upon this restaurant on my way home from the subway." can be expressed as "NN -1 NN -1 NP -1", but the internal structure of the phrase represented by "NP" in the construction is very different, and the same sentence construction is used to represent different sentences to guide the two The composition of phrases or sentences is irrational. Where the words in the sentence "Not only did they have amazing, sandwiches, soup, pizza, etc, but their homemade sorbets are out of this world!" are "soup" (NN), "pizza" (NN), and "world" "(NP). In the sentence "I stumbled upon this restaurant on my way home from the subway." "NN" is "restaurant", 'way', "home"; "NP" is "subway", the rest are "-1" ” contains punctuation. Therefore, it is necessary to obtain the internal structure information of the words in the sentence to improve the ability of the construction information to distinguish the basic structure of the sentence.

Point 4: Figure 1 (a): The organization of the network is similar to the Sentence-BERT, which has a Siamese-network topology (Figure 1 and 2 of [10]), but these networks are used when training the Sentence-BERT using the STS task that compares two sentences. The Sentence-BERT itself is not a Siamese network but a BERT model with pooling. On the other hand, the proposed CharAs-CBert network looks to receive one sentence and determines its sentiment. Why the proposed model takes two sentences?

Response 4: Thank you very much for the experts' opinions. We have made detailed modifications to this problem. 

Point 5: Section 3.1: It is explained that the proposed network first takes words, characters, and constructions of a sentence and embed them using a pre-trained BERT. However, a usual BERT model only takes q sequence of words (or tokens) as an input. How the authors input characters and constructions into the BERT?

Response 5: Thank you very much for the experts' opinions. We have made detailed modifications to this problem. 

For instance,

Characters are the smallest units that make up a word, and the adjacent relationship between them reflects the internal structural information of the word. A sentence usually contains multiple constructions, and the constructions of different sentences may also be the same. When the constructions of different sentences are the same, it is not conducive to distinguishing between different sentences. Therefore, obtaining character information of words in a sentence is helpful for sentences. The distinction between representation and basic structure. For example, the subjective attitude of the sentence "We would return to this place again!" is "positive", and the subjective attitude of the sentence "Too bad the food wasn't of the same heritage." is "negative", but they have the same The construction "DT -1 NN -1 . -1", at the same time, there are also great differences in the length and composition of the phrases corresponding to "DT" and "NN$" in the construction, and it may be better to use character information to distinguish these phrases from the inside. efficient. Where "DT" means determiner; "NN" means Noun, singular or mass, etc. Therefore, we construct a topological graph of a character and use graph convolution [26] to optimize and adjust this topological graph, namely the character graph convolution module (CharGCM), to ensure the accuracy of the internal structure information of keywords.

Point 6: Line 140: The abbreviated word BSL is not defined elsewhere.

Response 6: Thank you very much for the experts' opinions. We have made detailed modifications to this problem. 

For instance,

Zhang et al. [22] proposed an unsupervised sentence representation learning method based on Bootstrapped Sentence Representation Learning (BSL). Given an enhanced view of each sentence, the sentence was trained to adjust a branch of the network online, among other different sentence representations. This method is obviously due to other unsupervised methods, but it still has some shortcomings in sentence enhancement.

Point 7: Table 2: Results of many conventional(?) methods are shown here, but none of those methods are explained. Thus, I could judge that the proposed method is truly better than the other methods. 

The experimental result for the same dataset is shown in the literature A) above, which was just published by the same authors. I noticed that the absolute performance is different. It is natural that the methods are different, but I wonder why these two papers employed completely different set of recognition methods for the same task.

Response 7: Thank you very much for the experts' opinions. We have made detailed modifications to this problem. 

For instance,

According to Table 1, we can draw the following conclusions:

(1) The overall performance of our proposed CharAs-CBert sentence representation framework on the three baseline datasets outperforms other representation models, such as F1 in Laptop, Restaurant, and ACL14 than SBert-att 1.1%, 1.03%, and 1.17% respectively. There are three possible reasons. First, we use the Slice Attention Module (SAM) to establish long-term dependent salient word representations from two directions, such as channel and space. The performance of sentence representation; second, BIRM and CharGCM are introduced to support construction information, explore the internal structure information of sentences, and highlight the differences between different sentences, resulting in the improvement of sentence representation performance; third, the fusion of Three different feature vectors make up for the insufficiency of a single representation, and understand sentences from different angles and levels. In addition, the introduction of rich low-level semantics further enhances the difference between sentences, improves the performance of sentence representation, and improves the downstream emotion. Accuracy for classification tasks.

(2) Compared with Bert-LSTM and Tree-LSTM sentence representation models, TG-HTreeLSTM and TE-DCNN have certain competitive advantages in three types of data. For example, on the Laptop data, the F1 of TG-HTreeLSTM is 6.53% higher than that of Tree-LSTM. The possible reason is that Tree-LSTM can only process binary selection trees that are different from the original selection tree. Conversely, TG-HTreeLSTM can process the original constituency tree of sentences, resulting in a performance improvement. The good performance of TE-DCNN may be because its dynamic synthesis strategy plays an important role, resulting in better semantic information obtained by the network.

(3) Capsule-B is improved by 0.98%, 1.42%, and 2.01% respectively compared with F1 of TE-DCNN. The possible reason for this is that the capsule network can perceive more effectively due to the directionality of capsule neurons. The subtle changes between different sentences improve the distinguishing ability of sentence structure, thereby improving the representation effect of sentences.

(4) On the ACL14 baseline data, the ACC of CNN-LSTM is 1.09% higher than that of the LSTM method. The possible reason is that CNN obtains the local spatial features of sentences, LSTM encodes the time series, and establishes a complementary relationship between the spatial and temporal features. Improved sentence representation. Thus, the sentence representation accuracy is improved. In contrast, Self-Att achieves better competitive advantages in three sets of open-source baseline datasets, mainly since self-attention focuses on key information and effectively models the local and global semantics of sentences.

Point 8: Section 3.3: One of motivations to introduce BIRM looks like that the tanh activation could be improved by ReLU activation. Changing the activation of LSTM is examined in the following paper:

  1. B)      Kent, D., & Salem, F. (2019, August). Performance of three slim variants of the long short-term memory (LSTM) layer. In 2019 IEEE 62nd International midwest Symposium on circuits and Systems (MWSCAS) (pp. 307-310).

According to this work, changing the activation function from tanh did not improve the performance. Thus, the authors need to validate whether changing the activation improved the result compared with BLSTM. Table 2 shows comparison with many other models, but it lacks viewpoints from which the authors prove that the changes employed in the proposed method were effective.

Response 8: Thank you very much for the experts' opinions. We have made detailed modifications to this problem. 

For instance,

According to Table 3, we draw the following conclusions.

(1) Compared with the single-structure sentence representation, the multi-feature co-representation method shows better performance. Such as CharAs-CBert(SAM+CharGCM) vs CharAs-CBert(CharGCM), CharAs-CBert(BIRM) and CharAs-CBert(SAM) F  on ACL14 baseline datasets increased by 1.27%, 1.78% and 2.02% respectively. The possible reason is that the multi-feature vector fusion understands the sentence from different angles, and the different feature vectors form complementarity, making up for a single feature vector that is easy to ignore A question of detail semantics. In addition, the CharAs-CBert(BILSTM) method is inferior to the CharAs-CBert(BIRM) method on the three sets of open-source baseline data, which indicates that the proposed BIRM plays a positive role in the overall performance of the model, possibly The reason is that stacking multiple layers of bidirectional independent recurrent neural networks obtains a better global representation.

(2) On the Laptop baseline data, CharAs-CBert(Att+BIRM+CharGCM) is better than CharAs-CBert(SAM+RNN+CharGCM) and CharAs-CBert(SAM+BIRM+CharCNN) F is improved by 0.1% and 0.27%, respectively, which shows that our proposed components play a positive role in the overall performance of the model. In addition, we also found that the SAM component has the least positive effect on the model. It may be that the model only uses the word vector to represent the sentence in the absence of construction, ignoring the basic structure of the sentence, and cannot fully obtain the context of the sentence. Semantic details.

(3) Although SAM+BILSTM+CharGCM has achieved a certain competitive advantage, it is still lower than CharAs-Bert. Building deep BILSTMs for learning key semantics in data is difficult. In contrast, BIRM can be stacked into very deep networks using non-saturating activation functions, etc. We obtain better depth semantics and perceive richer detail changes due to stacking in the form of residuals.

Reviewer 2 Report

This manuscript proposed a framework for character-assisted construction of Bert sentences to explore the global semantics of sentences from different perspectives and contexts. The proposed framework was tested using different data sets and achieved good results. This work is a good study case and suitable for the Journal scope and reader. However, the technical writing lacks organization and has many grammatic errors, making it hard to read. Test results were not discussed in detail and needed more comprehensive evaluations and comparisons with other researchers to validate the results. The paper needs to be restructured entirely and revised before publication.

In addition to the following comments:

- The abstract is broad and does not emphasize this work's objectives, methodology, and contributions. It lacks quantitative results and a summary of findings.

- Enhance the introduction to present a detailed background related to the current problem and solutions.

- Add more recent literature (2022-2021) that describes the works developed in the last few years and then conclude the advantages and disadvantages of each method in a tabular form.

-What are the current research gaps in the studies mentioned in the literature survey, and how will this work fill them?

-The conclusion should present the current work's new findings and numerical results and explain the limitations. Also, add to the future directions.

- Compare the obtained results with other researchers (mentioned in the related work) using different evaluation factors supported by graphical and tabular data? 

- Avoid using I, WE, OUR, etc., in the paper context.

- Do not use short terms without a prior definition (K-NN, SVM, CNN, LSTM, BSL), etc.

-Too-long sentences make the meaning unclear. Consider breaking it into multiple sentences—for example, L5-L8; L10-L12; L24-L27; L451-L456;  etc.

-Many grammatical or spelling errors that make the meaning unclear and sentence construction errors need proofreading. Improve the English language, redaction, and punctuation in general. The manuscript should undergo editing before being submitted to the journal again. 

The following are some examples:

L6: the sentence, so as to reduce   ... should be ... the sentence to reduce 

L7: sentences, and at the same time, ... should be ... sentences. At the same time,

L13: various open source baseline data... should be ... various open-source baseline data

L26: The basic structure of ... should be ... the basic structure of

Author Response

Point 1:  The abstract is broad and does not emphasize this work's objectives, methodology, and contributions. It lacks quantitative results and a summary of findings.

Response 1: Thank you very much for the experts' opinions. We have made detailed modifications to this problem. 

For instance, In the process of semantic capture, traditional sentence representation methods tend to lose a lot of global and contextual semantics and ignore the internal structure information of words in sentences. To address these limitations, we propose a sentence representation method for character-assisted construction-Bert (CharAs-CBert) to improve the accuracy of sentiment text classification. First, based on the construction, a more effective construction vector is generated to distinguish the basic morphology of the sentence, reduce the ambiguity of the same word in different sentences, and at the same time, strengthen the representation of salient words and effectively capture contextual semantics. Second, characters feature vectors are introduced to explore the internal structure information of sentences and improve the representation ability of local and global semantics. Then, to make the sentence representation have better stability and robustness, characters information, word information, and construction vectors are combined and used together for sentence representation. Finally, the evaluation and verification are carried out on various open-source baseline data such as ACL-14 and SemEval 2014 to demonstrate the validity and reliability of sentence representation, namely, the F1 and ACC are 87.54\% and 92.88\% on ACL14, respectively.

Point 2: Enhance the introduction to present a detailed background related to the current problem and solutions.

Response 2:Thank you very much for the experts' opinions. We have made detailed modifications to this problem. 

For instance,

Unlike convolutional neural networks [6], which are limited by the receptive field, the attention mechanism [7] balances the interrelationships between words in a sentence using weight calculation and assignment, meanwhile, highlighting the representation of salient features. Recurrent neural networks cannot establish effective long-term dependencies in more complex sentence representation tasks due to problems such as gradient disappearance and explosion. Independent recurrent neural network (IndRNN) [9] improves neuron performance because each neuron has independent spatiotemporal features. The interpretability of the input behavior, and the cross-layer connection between them, prevents the gradient from disappearing and exploding and promotes the long-term learning of the network to improve the modeling ability of dependencies. In the traditional language model, Bert needs a lot of time for clustering and similarity analysis of sentence pairs to realize the embedded representation of sentences. For example, the sentence-BERT [10] algorithm uses the average value of the BERT output vector for supervised learning to achieve efficient sentence embedding. But when interpreting sentences, the same word may also be ambiguous in different sentences, that is, the role of a word in a sentence depends on the context of the sentence, not entirely on the word itself. In addition, when two sentences have the same construction, the phrases that make up the sentence may also be quite different. Such constructions as "Not only did they have amazing, sandwiches, soup, pizza, etc, but their homemade sorbets are out of this world!" and "I stumbled upon this restaurant on my way home from the subway." can be expressed as "NN -1 NN -1 NP -1", but the internal structure of the phrase represented by "NP" in the construction is very different, and the same sentence construction is used to represent different sentences to guide the two The composition of phrases or sentences is irrational. Where the words in the sentence "Not only did they have amazing, sandwiches, soup, pizza, etc, but their homemade sorbets are out of this world!" are "soup" (NN), "pizza" (NN), and "world" "(NP). In the sentence "I stumbled upon this restaurant on my way home from the subway." "NN" is "restaurant", 'way', "home"; "NP" is "subway", the rest are "-1" ” contains punctuation. Therefore, it is necessary to obtain the internal structure information of the words in the sentence to improve the ability of the construction information to distinguish the basic structure of the sentence.

To alleviate the ambiguity of the same word in different sentences and combine the construction to explore the internal structure information of the sentence, a better sentence representation can be obtained. We propose a sentence representation framework of character-assisted construction Bert to improve the representation of global and contextual semantic information in sentiment texts. The approach has three main contributions:

When understanding a sentence, the weight of the word-processed by Bert is not directly used to explain the sentence, but a slice attention enhancement network is designed to explain these behaviors, assigning salient words in the sentence to the sentence. Higher weight coefficients, meanwhile, explore the channel dependencies and spatial correlations of different salient words in the sentence.

Based on our sentence construction, we design a bidirectional independent recurrent neural network to explore the construction vector of sentences, alleviate the ambiguity of the same word in different sentences, and promote long-term learning of the network to establish effective long-term dependencies, it realizes the interaction between forward and backward semantic information and improves the model's ability to perceive contextual details.

A construction-based character graph convolutional network is designed to explore the internal structural information of salient words in sentences, that is, there is a strong correlation between adjacent characters in each salient word. Features strengthen construction information to improve the ability of construction information to distinguish the basic structure of sentences. Furthermore, we design a triple loss function to better tune and optimize the network to learn better sentence

Point 3: Add more recent literature (2022-2021) that describes the works developed in the last few years and then conclude the advantages and disadvantages of each method in a tabular form.

Response 3: Thank you very much for the experts' opinions. We have made detailed modifications to this problem. 

For instance,

With the development of the machine learning model, the accuracy of the sentence representation model is gradually improved, but there is still the problem of incomplete sentence semantic expression. With the emergence of deep learning, the unique learning model of the neural network provides a distributed lexical expression for sentences, which further promotes the development of this field. Ma J et al. [16] proposed a tree-dependent self-attention network by combining the self-attention mechanism with trees, considering the basic form of sentences, to explore the deep semantic information of sentences, but the network was not able to obtain features when acquiring features, relying too much on the basic structural information of the sentence, ignoring the role of some keywords in the sentence. Task-Related Sentence Representation Bai X et al. [17] designed a neighbor attention sentence representation method, which explicitly pointed out the label space in the input and predicted the class of the label by adding mask labels while using the fusion label to obtain the sentence's semantic information. But the model is too complex and relies heavily on label information. Hu X et al. [18] proposed a pruned trained recurrent neural network for grammar induction and text representation using a top-down parser as a pruning method for the model and parallel encoding during inference, but this method in the pruning process is easy to ignore the detailed semantics, which leads to the reduction of the representation accuracy of the model. Fu et al. [6] proposed a coding-decoder model with an attention mechanism. Convolution neural networks (CNN) were used as an encoder to learn distributed sentence representation features, and Long Short-Term Memory (LSTM) with an attention mechanism as a decoder could not only reconstruct the original sentence structure but also multi-task to predict the next sentence content. Good results are obtained in multiple datasets. Zhao et al. [19] use the attention mechanism to obtain semantic representation at different levels in sentences, which can more accurately express the emotion reflected in the text and make the sentence representation more comprehensive. 

Point 4: What are the current research gaps in the studies mentioned in the literature survey, and how will this work fill them?

Response 4: Thank you very much for the experts' opinions. We have made detailed modifications to this problem. 

Point 5: The conclusion should present the current work's new findings and numerical results and explain the limitations. Also, add to the future directions.

Response 5: Thank you very much for the experts' opinions. We have made detailed modifications to this problem. 

For instance,

In this paper, we design a new character-assisted structure Bert sentence representation framework, which utilizes words, structures, and characters to explore the context and global semantics of sentences from different perspectives and helps to capture the sentence's meaning based on structure and character information. The internal structure information improves the ability to distinguish between different sentences, and at the same time, due to the complementary and interactive relationship between different feature vectors, the ambiguity of the same word in different sentences is reduced. Finally, the evaluation results on baseline data such as ACL14 and SemEval 2014 show that the proposed CharAs-CBert sentence representation framework has good robustness and effectiveness, that is, the experimental results on different baseline datasets are superior to other sentence display methods.

Although the proposed CharAs-CBert sentence representation framework has achieved good representation performance, we found that this method still has shortcomings during the experiment, such as the complex model structure, and different feature vectors having some redundant information in the fusion stage. At the same time, there is still a lot of room for improvement in time efficiency. Therefore, in the following, we will design an efficient and concise semantic-guided sentence representation framework from the perspective of model structure and exploration of more effective detail semantics, to obtain semantic details while ensuring accuracy and improving model efficiency.

Point 6: Compare the obtained results with other researchers (mentioned in the related work) using different evaluation factors supported by graphical and tabular data? 

Response 6: Thank you very much for the experts' opinions. We have made detailed modifications to this problem. 

For instance,

According to Table 1, we can draw the following conclusions:

(1) The overall performance of our proposed CharAs-CBert sentence representation framework on the three baseline datasets outperforms other representation models, such as F1 in Laptop, Restaurant, and ACL14 than SBert-att 1.1%, 1.03% and 1.17% respectively. There are three possible reasons. First, we use the Slice Attention Module (SAM) to establish long-term dependent salient word representations from two directions, such as channel and space. The performance of sentence representation; second, BIRM and CharGCM are introduced to support construction information, explore the internal structure information of sentences, and highlight the differences between different sentences, resulting in the improvement of sentence representation performance; third, the fusion of Three different feature vectors make up for the insufficiency of a single representation, and understand sentences from different angles and levels. In addition, the introduction of rich low-level semantics further enhances the difference between sentences, improves the performance of sentence representation, and improves the downstream emotion. Accuracy for classification tasks.

(2) Compared with Bert-LSTM and Tree-LSTM sentence representation models, TG-HTreeLSTM and TE-DCNN have certain competitive advantages in three types of data. For example, on the Laptop data, the F1 of TG-HTreeLSTM is 6.53% higher than that of Tree-LSTM. The possible reason is that Tree-LSTM can only process binary selection trees that are different from the original selection tree. Conversely, TG-HTreeLSTM can process the original constituency tree of sentences, resulting in a performance improvement. The good performance of TE-DCNN may be because its dynamic synthesis strategy plays an important role, resulting in better semantic information obtained by the network.

(3) Capsule-B is improved by 0.98%, 1.42%, and 2.01% respectively compared with F1 of TE-DCNN. The possible reason for this is that the capsule network can perceive more effectively due to the directionality of capsule neurons. The subtle changes between different sentences improve the distinguishing ability of sentence structure, thereby improving the representation effect of sentences.

(4) On the ACL14 baseline data, the ACC of CNN-LSTM is 1.09% higher than that of the LSTM method. The possible reason is that CNN obtains the local spatial features of sentences, LSTM encodes the time series and establishes a complementary relationship between the spatial and temporal features. Improved sentence representation. Thus, the sentence representation accuracy is improved. In contrast, Self-Att achieves better competitive advantages in three sets of open-source baseline datasets, mainly since self-attention focuses on key information and effectively models the local and global semantics of sentences.

Point 7: Avoid using I, WE, OUR, etc., in the paper context.

Response 7: Thank you very much for the experts' opinions. We have made detailed modifications to this problem. 

For instance,

To alleviate the ambiguity of the same word in different sentences and combine the construction to explore the internal structure information of the sentence, a better sentence representation can be obtained. We propose a sentence representation framework of character-assisted construction Bert to improve the representation of global and contextual semantic information in sentiment texts. The approach has three main contributions:

When understanding a sentence, the weight of the word-processed by Bert is not directly used to explain the sentence, but a slice attention enhancement network is designed to explain these behaviors, assigning salient words in the sentence to the sentence. Higher weight coefficients, at the same time, explore the channel dependencies and spatial correlations of different salient words in the sentence.

Point 8: Do not use short terms without a prior definition (K-NN, SVM, CNN, LSTM, BSL), etc.

Response 8: Thank you very much for the experts' opinions. We have made detailed modifications to this problem. 

For instance,

For several traditional machine learning algorithms, Bayhaqy et al. [11] respectively used the methods of the decision tree, K-Nearest Neighbor (K-NN), and Naive Bayes classifier to conduct experiments on the data of different users' opinions on the Indonesian market on Twitter. Naive Bayes classifiers are more suitable for sentiment analysis in the market research category. Rathi et al. [12] analyzed various machine learning algorithms on Twitter comments, but the effect was not good. Finally, they proposed the method of combining support vector machine (SVM) and decision trees through experiments. Zuo et al. [13] conducted emotion analysis on large-scale Steam Review data set, respectively using a fully supervised machine learning decision tree and naive Bayes. higher than the naive Bayes algorithm, which benefited from the additional features of the data set to enhance the learning effect of the decision tree. Chang et al. [14] wrote the use of library support vector machine (LIBSVM) of support vector machine (SVM) and gave many details of support vector machine (SVM) implementation so that more people can use SVM to realize sentence classification more conveniently. Li et al. [15] based on library support vector machine (LIBSVM) library and Gaussian radial basis function kernel with kernel and cost parameters as default values experimented with support vector machine (SVM) and nearest Neighbor algorithm (NN) to achieve emotion classification.

Point 9: Too-long sentences make the meaning unclear. Consider breaking it into multiple sentences—for example, L5-L8; L10-L12; L24-L27; L451-L456; etc.

Response 9: Thank you very much for the experts' opinions. We have made detailed modifications to this problem. 

For instance,

In addition, when two sentences have the same construction, the phrases that make up the sentence may also be quite different. Such constructions as "Not only did they have amazing, sandwiches, soup, pizza, etc, but their homemade sorbets are out of this world!" and "I stumbled upon this restaurant on my way home from the subway." can be expressed as "NN -1 NN -1 NP -1", but the internal structure of the phrase represented by "NP" in the construction is very different, and the same sentence construction is used to represent different sentences to guide the two The composition of phrases or sentences is irrational. Where the words in the sentence "Not only did they have amazing, sandwiches, soup, pizza, etc, but their homemade sorbets are out of this world!" are "soup" (NN), "pizza" (NN), and "world" "(NP). In the sentence "I stumbled upon this restaurant on my way home from the subway." "NN" is "restaurant", 'way', "home"; "NP" is "subway", the rest are "-1" ” contains punctuation. Therefore, it is necessary to obtain the internal structure information of the words in the sentence to improve the ability of the construction information to distinguish the basic structure of the sentence.

Point 10: Many grammatical or spelling errors that make the meaning unclear and sentence construction errors need proofreading. Improve the English language, redaction, and punctuation in general. The manuscript should undergo editing before being submitted to the journal again. 

The following are some examples:

L6: the sentence, so as to reduce   ... should be ... the sentence to reduce 

L7: sentences, and at the same time, ... should be ... sentences. At the same time,

Response 10: Thank you very much for the experts' opinions. We have made detailed modifications to this problem. 

For instance,

Considering that in the process of sentence extraction, RNN is prone to problems such as gradient disappearance and explosion. At the same time, it is difficult to establish effective long-term dependencies, and it is difficult to explain the input behavior due to the entanglement of neurons. Although LSTM [9] can effectively solve a series of problems existing in RNNs [9], the use of hyperbolic tangent and sigmoid activation functions leads to hierarchical gradient decay, which cannot effectively capture the detailed features of sentences. Therefore, to address these limitations for more efficient sentence representations, we design a bidirectional independent recurrent neural network module (BIRM), which models the construction information from both forward and backward directions And realize the interaction of different directional features, which helps to obtain global and contextual details. In addition, the neurons in the same layer in this module are independent of each other, which is beneficial to the interpretation of input behavior. At the same time, the information flow is realized between different layers. Cross-layer connections help to model detail semantics and better transfer detail semantics to lower layers. The specific description of BIRM is as follows.

L13: various open source baseline data... should be ... various open-source baseline data

Response 12: Thank you very much for the experts' opinions. We have made detailed modifications to this problem. 

For instance,

To demonstrate the effectiveness of the proposed CharAs-CBert sentence representation framework, we conducted evaluation and verification on open-source baseline data such as SemEval2014 and ACL14. Table 2 shows the experimental results of different sentence representation methods.

L26: The basic structure of ... should be ... the basic structure of

Response 13: Thank you very much for the experts' opinions. We have made detailed modifications to this problem. 

For instance,

A construction-based character graph convolutional network is designed to explore the internal structural information of salient words in sentences, that is, there is a strong correlation between adjacent characters in each salient word. Features strengthen construction information to improve the ability of construction information to distinguish the basic structure of sentences. Furthermore, we design a triple loss function to better tune and optimize the network to learn better sentence representations.

This module explores the global and contextual semantics of sentences from different perspectives and levels by obtaining word vectors, character vectors, and construction vectors of emotional sentences. The character vector and construction vector can start from the internal structure of the words in the sentence and the basic structure of the sentence, effectively reducing the ambiguity of the same word in different sentences. meanwhile, it is helpful for sentence representation.

As one of the important operations of natural language processing, sentence embedding representation is widely used in many tasks such as text classification [1], semantic matching [2], machine translation [3], and knowledge question answering [4]. The current popular sentence representation methods are mainly based on neural networks and pre-trained language models. The most widely used neural network models are long and short-term memory networks [5], as well as convolution [6] and attention models [7], etc. When processing a sentence, these neural network methods are in a smooth order, while the basic structure of the sentence is not considered, meanwhile, in the subsequent sentence synthesis process, the basic syntactic information of the sentence is ignored, such as the obvious difference between the synthesis of "adverb-noun" and "adjective-noun". When using a simple pre-trained language model BERT \cite{ref-8} to achieve sentence representation, it is easy to lose a lot of sentence details, resulting in sentence representation tasks that are still lower than the traditional Glove word embedding representation method. Therefore, the Bert language model is usually combined with neural network methods. Improving sentence representation has become a new trend.

Round 2

Reviewer 1 Report

1.       On the use of “BERT” and “Bert”: It is OK to use “CharAs-CBert” because this name is proposed by the authors. However, there are still some confusions between “BERT” and “Bert” in the body text. For example, the notation “Bert” remains at line 31, 43, 66, 68, 177, 304, 492, and the caption of Figure 1.

2.       On the “sentence construction”: On line 202-204, the authors added some descriptions on the sentence construction. However, the “construction” used here does not look like a result of construction grammar at all. The “construction” is not a tree but a sequence of symbols. The authors need to explain what the “construction” is, and how did the authors derive the construction from a given sentence.

3.       On the correspondence between the construction and the sentence: I do not understand how the construction is calculated from a sentence. The authors need to explain what NN, NP, or CC means, too. For example, in the example shown in the revised part, “Not only did they have amazing, sandwiches, soup, pizza, etc, but their homemade sorbets are out of this world!” and “NN -1 NN -1 NP -1”, the explanation says soup=NN, pizza=NN, and world=NP. Why do we miss sandwiches and sorbets? Why the first two are NN and the last one NP? Why do we have no -1 before the first NN? I do not understand them at all.

4.       Figure 1: In the reply letter, the authors state that “We have made detailed modifications to this problem,” but the figure and its explanation do not change at all.

5.       On the “Point 5” in the reply letter: The question I posed is how the character and construction information is processed by a pre-trained BERT, which only takes words (or tokens). The reply and the revised description do not answer this question at all.

6.       Line 122: the citation of Bai X et al. is not numbered (displayed as [?])

7.       There are still some typos and grammatical mistakes (Not limited to the following examples). The manuscript should be proofread.
Line 68: “the weight of the word-processed by Bert” -> “the weights of the words processed by BERT”
Line 90: “Relate work” -> “Related works”
Line 98: “classifier” -> “classifiers”
Line 99: “classifiers are” -> “classifier is”
Line 157: “usrl-MTL” -> “USR-MTL”
Caption of Figure 1: “C^0 and C^00 indicates” -> “C^0 and C^00 indicate”; “product operate” -> “product operation”; “shuffle operate” -> “shuffle operation”; “a forward layers” -> “forward layers”; “f_p^A and f_p^B represents” -> “f_p^A and f_p^B represent”
Line 178: “makeup” -> “make up”
Line 233: “meanwhile” -> “Meanwhile”
Line 244: “i.e.” -> “i.e.,”
Line 248-249: “operate” -> “operation”
Line 297: “inside. efficient.” -> “inside.”
Line 308: “et.” -> “etc.”
Line 333-335: “… and … represents” -> “… and … represent”
Caption of Table 3: “EExperimental” -> “Experimental”

Author Response

Point 1: On the use of “BERT” and “Bert”: It is OK to use “CharAs-CBert” because this name is proposed by the authors. However, there are still some confusions between “BERT” and “Bert” in the body text. For example, the notation “Bert” remains at line 31, 43, 66, 68, 177, 304, 492, and the caption of Figure 1.

Response 1: Thanks to the expert opinion, we have made detailed revisions to address these issues.

For instance,

In this paper, we design a new character-assisted structure BERT sentence representation framework, which utilizes words, structures, and characters to explore the context and global semantics of sentences from different perspectives and helps to capture the sentence's meaning based on structure and character information. The internal structure information improves the ability to distinguish between different sentences. At the same time, due to the complementary and interactive relationship between different feature vectors, the ambiguity of the same word in different sentences is reduced. Finally, the evaluation results on baseline data such as ACL14 and SemEval 2014 show that the proposed CharAs-CBert sentence representation framework has good robustness and effectiveness, that is, the experimental results on different baseline datasets are superior to other sentence display methods.

Point 2:  On the “sentence construction”: On line 202-204, the authors added some descriptions on the sentence construction. However, the “construction” used here does not look like a result of construction grammar at all. The “construction” is not a tree but a sequence of symbols. The authors need to explain what the “construction” is, and how did the authors derive the construction from a given sentence.

Response 2: Thanks to the expert opinion, we have made detailed revisions to address these issues.

For instance,

It is worth noting that "construction" usually means "construction grammar", which is a grammar theory that gradually emerged in the late 1980s and a research method and school adapted to almost the entire language category. Constructive grammar is born out of cognitive grammar, which is a rebellion against a formal grammar. It belongs to the category of cognitive linguistics in essence, but it has the characteristics of being an independent paradigm of language research. In a certain sense, constructionism has formed an independent school of research, whose purpose is to emphasize inductive descriptions to explain existing sentences, rather than to generate possible legal sentences according to rule constraints, a certain expression construction always corresponds to a certain meaning.

Point 3: On the correspondence between the construction and the sentence: I do not understand how the construction is calculated from a sentence. The authors need to explain what NN, NP, or CC means, too. For example, in the example shown in the revised part, “Not only did they have amazing, sandwiches, soup, pizza, etc, but their homemade sorbets are out of this world!” and “NN -1 NN -1 NP -1”, the explanation says soup=NN, pizza=NN, and world=NP. Why do we miss sandwiches and sorbets? Why the first two are NN and the last one NP? Why do we have no -1 before the first NN? I do not understand them at all.

Response 3: Thanks to the expert opinion, we have made detailed revisions to address these issues.

For instance,

we adopt an average weighting strategy count the words corresponding to "CC", "NP" and "DT" in the construction, and obtain the mean vector feature, where "CC" indicates coordinating conjunction; "DT" indicate determiner; "NP" indicate noun phrase or noun; in the sentence "But the staff was so horrible to us." "But" is used as a "connector (CC)" or "determiner (DT)" to the noun phrase "the staff" (NP) Limit the scope of the situation or connect with other words to form a specific state. It is worth noting that the words corresponding to "-1" in the construction are not calculated for the mean value. The character vector, word vector, and construction vector of the corresponding word in the sentence are shown in the equations.

Point 4:  Figure 1: In the reply letter, the authors state that “We have made detailed modifications to this problem,” but the figure and its explanation do not change at all.

Response 4: Thanks to the expert opinion, we have made detailed revisions to address these issues.

Point 5:  On the “Point 5” in the reply letter: The question I posed is how the character and construction information is processed by a pre-trained BERT, which only takes words (or tokens). The reply and the revised description do not answer this question at all.

Response 5: Thanks to the expert opinion, we have made detailed revisions to address these issues.

For instance,

A word is composed of several characters. We input the characters that make up the word into BERT for learning and training to obtain character features. Secondly, the construction of a sentence is also composed of specific words, and these specific words are input into BERT to get a vector feature. We think this vector feature is a construction vector that contains construction information.

Song Z, Xie Y, Huang W, et al. Classification of traditional chinese medicine cases based on character-level bert and deep learning[C]//2019 IEEE 8th Joint International Information Technology and Artificial Intelligence Conference (ITAIC). IEEE, 2019: 1383-1387.

Point 6:  Line 122: the citation of Bai X et al. is not numbered (displayed as [?])

Response 6: Thanks to the expert opinion, we have made detailed revisions to address these issues.

For instance,

Task-Related Sentence Representation Bai X et al. [17] designed a neighbor attention sentence representation method, which explicitly pointed out the label space in the input and predicted the class of the label by adding mask labels while using the fusion label to obtain the sentence's semantic information. But the model is too complex and relies heavily on label information. 

Point 7: There are still some typos and grammatical mistakes (Not limited to the following examples). The manuscript should be proofread.

(1) Line 68: “the weight of the word-processed by Bert” -> “the weights of the words processed by BERT”

Response 7: Thanks to the expert opinion, we have made detailed revisions to address these issues.

For instance,

When understanding a sentence, the weight of the word processed by BERT is not directly used to explain the sentence, but a slice attention enhancement network is designed to explain these behaviors, assigning salient words in the sentence to the sentence. Higher weight coefficients, meanwhile, explore the channel dependencies and spatial correlations of different salient words in the sentence.

Line 90: “Relate work” -> “Related works”

Response : Thanks to the expert opinion, we have made detailed revisions to address these issues.

For instance,  

We have changed the title of Section 2 "Relate work" to "Related works"

(2) Line 98: “classifier” -> “classifiers”

Response: Thanks to the expert opinion, we have made detailed revisions to address these issues.

For instance,  

For several traditional machine learning algorithms, Bayhaqy et al. [11] respectively used the methods of the decision tree, K-Nearest Neighbor (K-NN), and Naive Bayes classifiers to conduct experiments on the data of different users' opinions on the Indonesian market on Twitter.

(3) Line 99: “classifiers are” -> “classifier is”

Response: Thanks to the expert opinion, we have made detailed revisions to address these issues.

For instance,  

Naive Bayes classifiers is more suitable for sentiment analysis in the market research category.

(4) Line 157: “usrl-MTL” -> “USR-MTL”

Response: Thanks to the expert opinion, we have made detailed revisions to address these issues.

For instance,  

Xu et al.[23] also proposed an unsupervised multi-task framework (USR-MTL), which mixed multiple sentence learning tasks into the same framework and obtained more meaningful sentence representation input by learning word order, word prediction, and sentence order.

(5) Caption of Figure 1: “C^0 and C^00 indicates” -> “C^0 and C^00 indicate”; “product operate” -> “product operation”; “shuffle operate” -> “shuffle operation”; “a forward layers” -> “forward layers”; “f_p^A and f_p^B represents” -> “f_p^A and f_p^B represent”

Response: Thanks to the expert opinion, we have made detailed revisions to address these issues.

For instance, 

and  indicate the channels of input features and satisfies

(6) Line 178: “makeup” -> “make up”

Response: Thanks to the expert opinion, we have made detailed revisions to address these issues.

For instance,  

The CharAs-CBert sentence representation framework is mainly composed of three parts, namely BERT initial embedding module with character and construction information, the characters auxiliary module, and the downstream sentiment classification module. Among them, the Bert embedding module of characters and constructions aims to map words and the characters that make up words into a low-dimensional space according to the sentence construction, making it easier to represent sentences;

(7) Line 233: “meanwhile” -> “Meanwhile”

Response: Thanks to the expert opinion, we have made detailed revisions to address these issues.

For instance,  

This module explores the global and contextual semantics of sentences from different perspectives and levels by obtaining word vectors, character vectors, and construction vectors of emotional sentences. The character vector and construction vector can start from the internal structure of the words in the sentence and the basic structure of the sentence, effectively reducing the ambiguity of the same word in different sentences. Meanwhile, it is helpful for sentence representation.

(8) Line 244: “i.e.” -> “i.e.,”

Response: Thanks to the expert opinion, we have made detailed revisions to address these issues.

(9) Line 248-249: “operate” -> “operation”

Response: Thanks to the expert opinion, we have made detailed revisions to address these issues.

For instance,  

Where indicates the operation of channel attention;

(10) Line 297: “inside. efficient.” -> “inside.”

Response: Thanks to the expert opinion, we have made detailed revisions to address these issues.

For instance,  

it may be better to use character information to distinguish these phrases from the inside.

(11) Line 308: “et.” -> “etc.”

Response: Thanks to the expert opinion, we have made detailed revisions to address these issues.

For instance,  

The overall characters can be as "abcdefghijklmnopqrstuwxyzABCDEFGHIJ~+=-<>()*[]}" etc.

(12) Line 333-335: “… and … represents” -> “… and … represent”

Response: Thanks to the expert opinion, we have made detailed revisions to address these issues.

For instance,  

Among them,  and  represent the character vectors of sentences A and B, respectively;  and  represent the word vector features of sentences A and B, respectively;  and  represent the construction vector features of sentences A and B, respectively;

(13) Caption of Table 3: “EExperimental” -> “Experimental”

Response: Thanks to the expert opinion, we have made detailed revisions to address these issues.

For instance,  

Experimental results of different components. SemEval2014 including Laptop and Restaurant.

Reviewer 2 Report

The revised manuscript is enhanced to the level that could publish in the current form based on the editorial board's opinion.

Author Response

Thanks to the expert opinion.